# Comparison of Satellite Soil Moisture Products in Mongolia and Their Relation to Grassland Condition

**Oyudari Vova \*, Martin Kappas and Ammar Rafiei Emam**

Cartography, GIS and Remote Sensing Department, Institute of Geography, University of Göttingen, Goldschmidtstr Street 5, 37007 Göttingen, Germany; mkappas@gwdg.de (M.K.); arafiei@geo.uni-goettingen.de (A.R.E.)

\* Correspondence: oyudari.vova@geo.uni-goettingen.de; Tel.: +49-176-25398594

**Abstract:** Monitoring of soil moisture dynamics provides valuable information about grassland degradation, since soil moisture directly affects vegetation cover. While the Mongolian soil moisture monitoring network is limited to the urban and protected natural areas, remote sensing data can be used to determine the soil moisture status elsewhere. In this paper, we determine whether in situ and remotely sensed data in the unaccounted areas of Southwestern Mongolia are consistent with each other, by comparing Soil Moisture and Ocean Salinity (SMOS) first passive L-band satellite data with in situ measurements. To evaluate the soil moisture products, we calculated the temporal, seasonal, and monthly average soil moisture content. We corrected the bias of SMOS soil moisture (SM) data using the in situ measured soil moisture with both the simple ratio and gamma methods. We verified the bias-corrected SMOS data with Nash–Sutcliffe method. The comparison results suggest that bias correction (of the simple ratio and gamma methods) enhances the reliability of the SMOS data, resulting in a higher correlation coefficient. We then examined the correlation between SMOS and Normalized Difference Vegetation Index (NDVI) index in the various ecosystems. Analysis of the SMOS and in situ measured soil moisture data revealed that spatial soil moisture distribution matches the rainfall events in Southwestern Mongolia for the period 2010 to 2015. The results illustrate that the bias-corrected, monthly-averaged SMOS data has a high correlation with the monthly-averaged NDVI ($R^2 > 0.81$). Both NDVI and rainfall can be used as indicators for grassland monitoring in Mongolia. During 2015, we detected decreasing soil moisture in approximately 30% of the forest-steppe and steppe areas. We assume that the current ecosystem of land is changing rapidly from forest to steppe and also from steppe to desert. The rainfall rate is the most critical factor influencing the soil moisture storage capacity in this region. The collected SMOS data reflects in situ conditions, making it an option for grassland studies.

**Keywords:** SMOS; soil moisture; statistics methods; Nash–Sutcliffe; NDVI; precipitation

## 1. Introduction

Soil moisture (SM) is an essential indicator of the hydrologic cycle that can affect the vegetation growth, impacting both global agriculture and grassland condition [1,2]. These impacts significantly concern herders in Mongolia, who depend on the pastureland for their livelihood. Mongolia is located in the Silk Road Economic Belt and has a high amount of grassland, most of which is used for pastoral purposes, which makes up a significant amount of the economic activity there [3]. Soil moisture can be used to evaluate drought risk and grassland condition in these arid lands. Accurate soil moisture data is necessary for short and long-term monitoring of grassland development. One previous study determined that 90% of pastureland in Mongolia is vulnerable to land degradation and desertification, and that 72% of that total territory is degraded to some degree; slight, moderate,

severe, and severely degraded grassland occupies 23%, 26%, 18%, and 5% of the vulnerable pastureland, respectively [4]. Climate change and overgrazing have caused significant grassland degradation in semi-arid regions [5–7].

Multiple analyses have compared satellite SM data to in situ SM measurements [8–10]. As point measurements, in situ measurements cannot necessarily be applied to a large area, and are difficult to obtain in high-altitude areas where few ground stations are available. Remote sensing data is beneficial for retrieving spatial and temporal soil moisture measurements over large and mountainous areas. SM derived from remote sensing only applies to up to the first five centimeters of soil depth, depending on the soil characteristics [11]. de Beurs et al. and Wessels et al. have found that vegetation changes, increasing temperatures, decreasing rainfall, and larger livestock populations have led to droughts and grassland degradation [12,13]. Several remote sensing vegetation indices, e.g., Normalized Difference Vegetation Index (NDVI), Enhanced Vegetation Index (EVI), and Soil-Adjusted Vegetation Index (SAVI) are widely used to assess changes in vegetation [14–16].

The Soil Moisture and Ocean Salinity (SMOS) satellite, the first and most successful space mission dedicated to monitoring global soil moisture, was launched in November 2009 [17,18]. The passive L-band (1.4 GHz) radiometer installed on this satellite acquires data for the entire globe every three days, with a spatial resolution of approximately 44 km. The overpass ascending time is 6:00 am local time and descending overpass time is 12 hours later [17,19,20]. Several studies have found that the quality and reliability of the SMOS products is sufficient [21–24], validation of SMOS SM data have been successful, and comparative analysis between SMOS data and in situ SM measurements have demonstrated a strong correlation [25]. One study even suggested that there is no difference between using a SMOS derived time series and the daily average of in situ SM measurements [26]. Recent studies suggest that SMOS provides successful results for North America, Australia, and central Asia [27]. The Soil Water Index (SWI), determined from SMOS products, has also been found to be an appropriate tool for drought monitoring [28]. A development overview of soil moisture studies from satellite sensors and their characteristics are summarized in Appendix A.

Accurate SM data collection depends heavily on atmospheric circulation and the weather. Previous research indicates that precipitation changes affect SM variability [29]. The annual temperature in Mongolia has warmed by 2.1 °C from 1940 to 2007, which, according to climate forecasts, will continue to rise to 3.1 °C above 1940 levels by 2050, as defined by the Mongolia Assessment Report on Climate Change [30]. Many studies have attributed grassland degradation to climate and precipitation patterns like drought and winter precipitation [31,32]. Few studies, however, have compared satellite SM products with in situ measurements in Mongolia. The SMOS SM data has been found to be an affordable indication of soil moisture and vegetation conditions, as well as drought monitoring [33]. Further analysis is necessary to identify the importance of SM for the future status of grassland condition and the preservation of ecological functions. The objectives of this study are: (1) To examine the spatial distribution of SMOS, SM, and Moderate Resolution Imaging Spectroradiometer (MODIS) NDVI and their relationship, and (2) compare satellite SMOS SM data with in situ measured SM data to evaluate grassland condition between 2000 and 2015 in Southwestern Mongolia.

## 2. Material and Methods

### 2.1. Study Area

The study area is located in central-southwestern Mongolia, consisting of four provinces Arkhangai, Uvurkhangai, Bayankhongor, and Gobi-Altai (Figure 1). From north to south, the climatic zones, the ecological zones, and relief characteristics vary considerably. The winter monthly average temperature is between −20 °C and −21 °C. The summer monthly average temperature is between 16 °C and 17 °C. Most rainfall occurs between June and September, showing an uneven distribution throughout the year. The altitude of the study area ranges from sea level to 4000 m.a.s.l. (Khangai Mountain). Steppe lies in the northwest of the study area, whereas the southern part encompasses the Gobi desert and drier climate. The vegetation growing season is from April–September. The soil type

of these regions is mainly Kastanozems and Chernozems. The soils are frozen from late November to the middle of March. The desert area soil types are Calcisols, Solonetz, and Solonchak, while the northern part of the mountain areas have Phaeozems and Cambisols. The southern region is almost exclusively a desert landscape, where agriculture and pasture fodder for livestock herds is scarce. These regions are more vulnerable to the impacts of climate change and droughts.

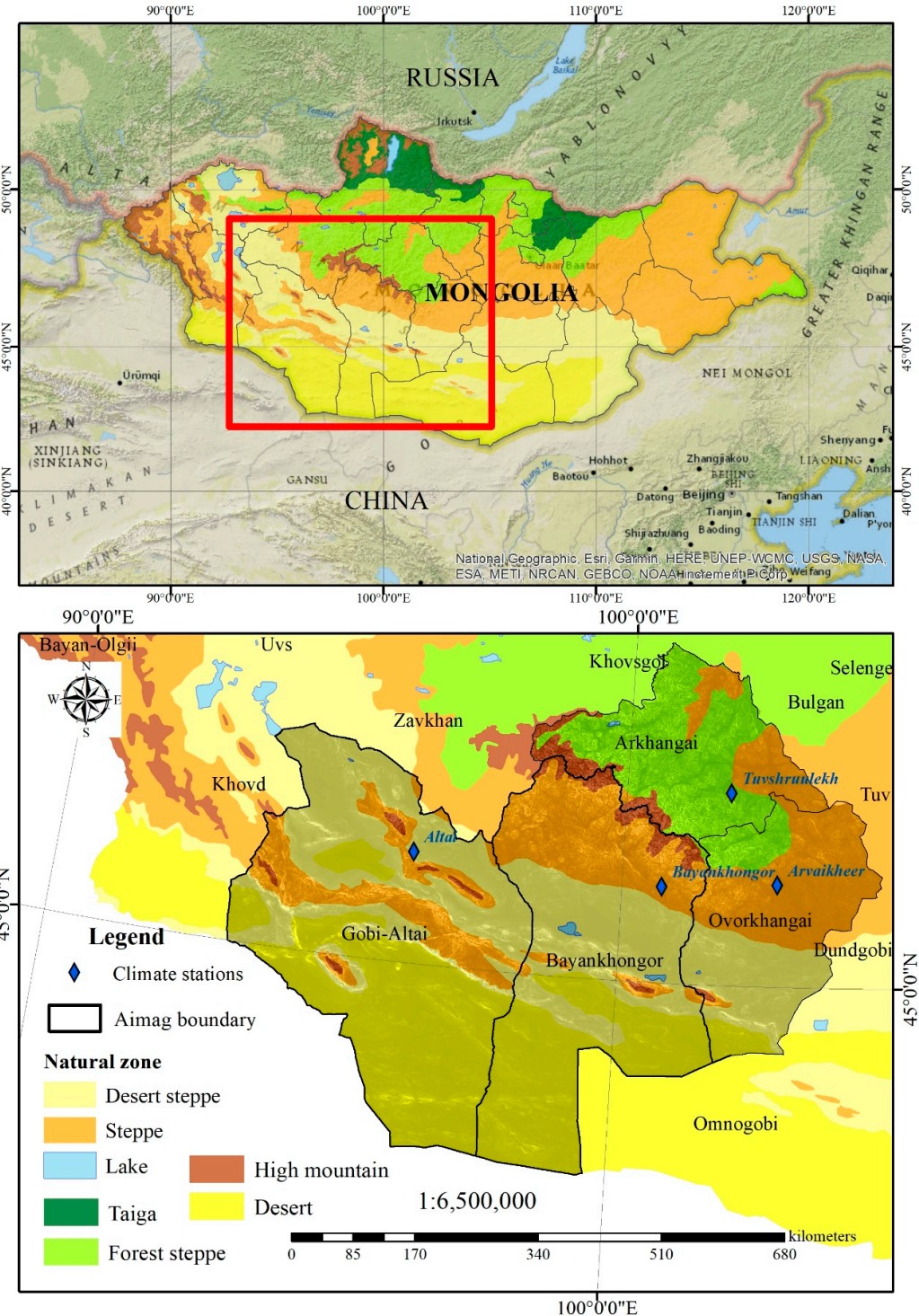

**Figure 1.** Meteorological stations of soil moisture (SM) measurements in Bayankhongor, Uvurkhangai, Arkhangai, and Gobi-Altai provinces, to represent each of the zone elevations. The figure shows the province boundaries and in situ SM measurement stations (blue color).

In this study, in situ SM measurements, precipitation data and soil map was obtained from the Institute of Meteorology and Hydrology of Mongolia.

The locations of the SM sampling stations have been categorized into four natural zones: forest-steppe, steppe, high mountain, and desert (Table 1). Nation-wide, the vegetation cover of the forest-steppe, steppe, and Gobi desert is 53%, 25%, and 15%, respectively [34]. The SM measurements were measured at vertical layers of soil from 10 to 15 cm depth.

**Table 1.** In situ measured SM data stations with information on location, elevation, natural zone, and soil types.

| Station Code | Station Name | Province Name | Lat (°N) | Long (°E) | Soil Type | Elevation (m) | Natural Zone |
|---|---|---|---|---|---|---|---|
| 287 | Bayankhongor | Bayankhongor | 46.192 | 100.718 | Kastanozems haplic | 1860 | Steppe |
| 277 | Altai | Gobi-Altai | 46.373 | 96.257 | Chernozems calcic, Calcisols haplic | 2147 | High mountain and desert |
| 288 | Arvaikheer | Uvurkhangai | 46.266 | 102.778 | Kastanozems haplic | 1831 | Steppe |
| 281 | Tuvshruulekh | Arkhangai | 47.388 | 101.906 | Kastanozems haplic | 1900 | Forest-steppe |

Figure 2 presents the soil type map of the study area. In situ SM measurements data has been taken from four different stations (provinces) and from the uncultivated land (natural rangeland). The southern region of the study area is desert steppe area, characterized by very sparse vegetation cover (shrubs), and the northern central region is more humid than the southern area and has forest-steppe and steppe.

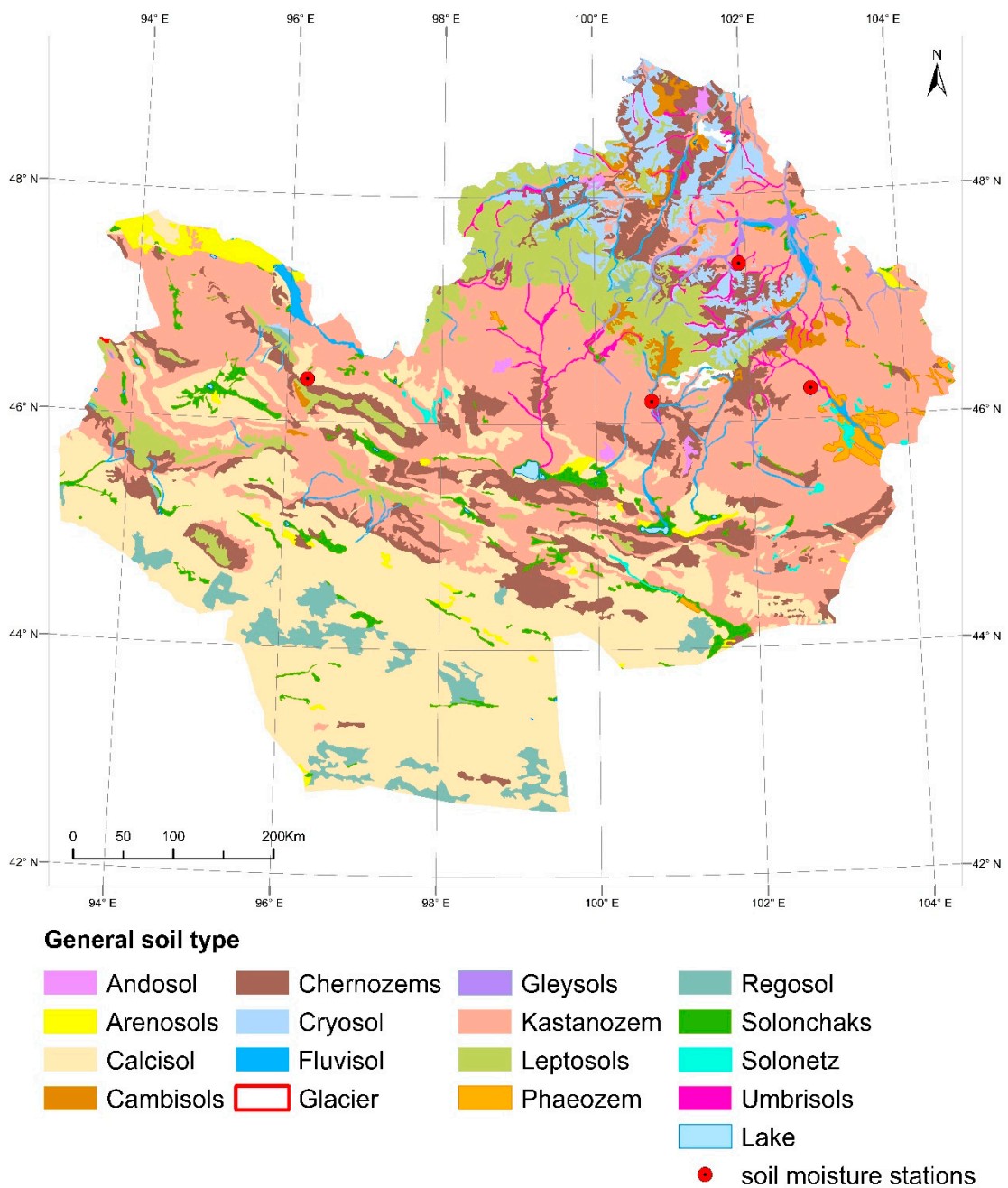

**Figure 2.** Overview of soil moisture stations and soil types in Southwestern Mongolia. Data is sourced from the Institute of Meteorology and Hydrology of Mongolia.

### 2.2. Methods

In this study, the SMOS L2 SM products were compared with in situ SM measurements. An area of 1822 × 684 km was examined, comprised of 918 SMOS L2 datasets. The statistical calculation of SM distributions of this region were studied from June to October to determine monthly averaged SMOS SM data for 2010–2015. These periods correspond to spring, summer, and autumn. We validated the SMOS SM data to assess the grassland conditions. We also examined the bias-corrected SMOS SM and NDVI data temporally and spatially from 2000 to 2015. Figure 3 presents a framework of the study that comprises pre-processing and GIS data processing, image processing, bias correction, and the relationship between SMOS SM, rainfall, and NDVI.

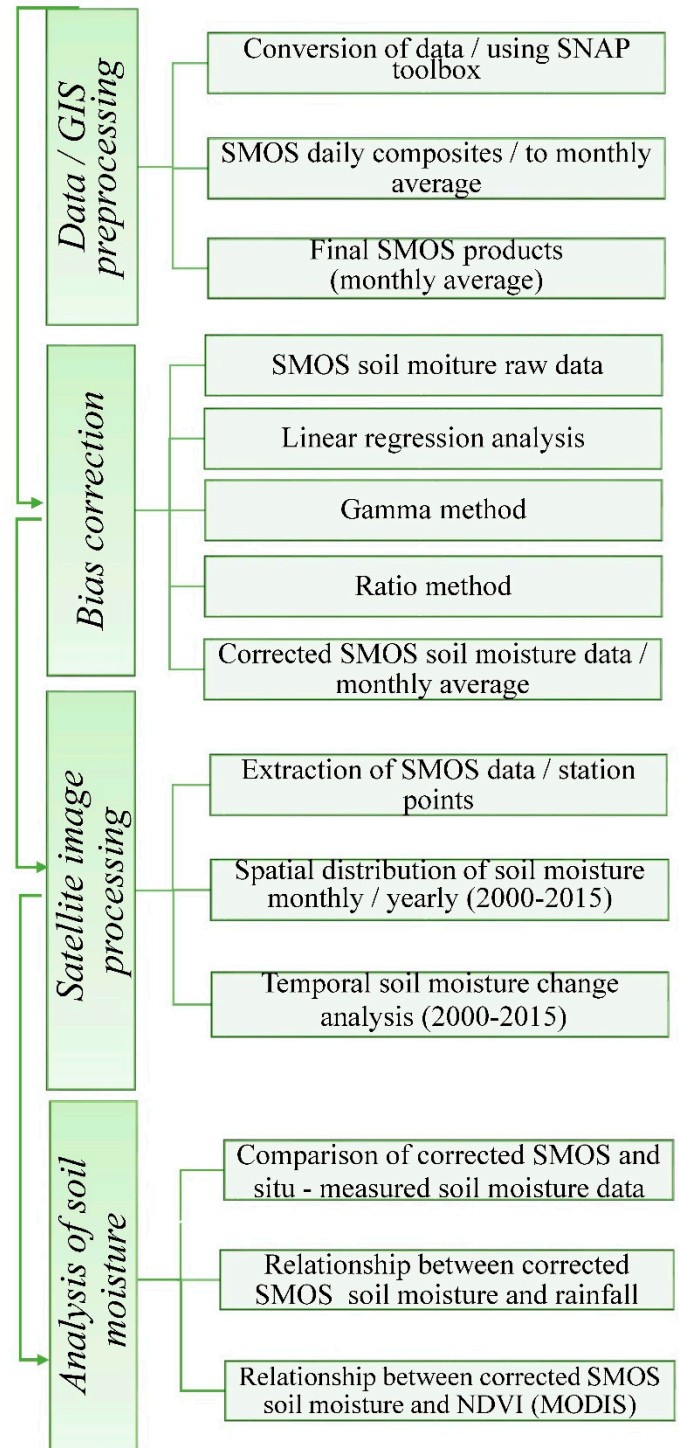

**Figure 3.** Flowchart of soil moisture evaluation steps.

2.2.1. Remote Sensing Data and Pre-processing

SMOS (SM Data)

The SMOS L2 data from January 2010 to December 2015 were selected for the comparison analysis and to determine the SM distribution. The SMOS data was extracted using the ArcGIS tool "Extract multi values to point." The data available are daily values, the monthly averages, and three parts per month. To accomplish further analysis, SMOS SM measurements were provided three times per

month (normally between the 29th and 8th, 9th and 18th, and 19th and 28th of every month) during the warm period of the year, which runs from April until the end of October. The radiometric SMOS SM data with an average spatial resolution of 43 km. Since the SMOS satellite data covers a large spatial area, it appears more reasonable to use the monthly average values. The datasets are provided in the netCDF format, ranging from regional to global scales and at a temporal resolution of three days. For checking the accuracy of the data, the bias-correction technique was used for different evaluation aspects. The bias-correction technique emphasizes the statistical characteristics of data and successfully reduces the error in data outputs, and has become very popular for correcting biases in multiple datasets and analyses [35,36].

In Situ Measured SM Data (Observation Dataset)

The National Agency of Meteorology, Hydrology and Environment Monitoring of Mongolia (NAMHEM) provided SM datasets from four different stations. In situ SM measurements used in this research were collected from 2010 to 2015. At each SM station, they collected one sample with the gravimetric method and converted the values to volumetric water content [37]. In situ measured SM data was obtained at a 10–15 cm depth at a monthly interval. For the majority of the stations, no data is available from April 8th and October 28th. For that reason, both these dates were excluded from the analysis.

MODIS NDVI Data (Vegetation Dataset)

The Moderate Resolution Imaging Spectroradiometer (MODIS) NDVI data used in this study are version 5 MODIS/Terra and MODIS/Aqua 1 km resolution daily daytime products (MOD13A2). These products are freely distributed by the U.S. Land Processes Distributed Active Archive Center [38–40]. The Normalized Difference Vegetation Index (NDVI) computed from space-borne observations at visible infrared wavelengths has been widely used since the 1980s to study the vegetation changes, soil, and drought [41]. The MODIS NDVI vegetation dataset used in this study is monthly averaged from 2010 to 2015. The NDVI was calculated using the following equation:

$$NDVI = (NIR - Red)/(NIR + Red),$$

where Red is the visible light of the red wavelength (from 400 - 700 nm) and NIR is the intensity of the near infrared wavelength (from 700–1100 nm). The spectral reflectance ratios indicate the reflected radiation over the incoming radiation in each spectral band, therefore these values range from 0.0 to 1.0. Individual NDVI pixel values were extracted from the images at each station location. For further statistical analysis, we resampled the MODIS NDVI data resolution to SMOS pixel size. At the regional scale, grassland degradation have been monitored in semi-arid regions using vegetation indices derived from remote sensing.

2.2.2. Processing of the Soil Moisture

Both the SMOS datasets and the in situ datasets were available in volumetric water content. Recently, bias correction method are often used for validation and comparison analysis. The bias correction technique significantly reduces the error in data output and emphasizes the statistical characteristics of observation data. Because of the coarse spatial resolution of the SMOS data, we applied two statistical approaches, the simple ratio method and the gamma method [35,36,42]. The simple ratio method dictates that the average SMOS SM for each month is divided by the corresponding in situ SM measurement. This factor is then multiplied by the daily SMOS SM data in order to receive a bias-free daily SM value. The gamma method is a bias-correction technique to decrease the biases in the SMOS SM data. The gamma distribution is a function of the probability density function (PDF) and the cumulative distribution function (CDF). More details about the bias technique can be found in [36]. Mishra et al. (2018) applied the bias correction technique to minimize the biases in the GCM

precipitation data [36]. In substance, the two-parameter gamma distribution was employed for bias correction. Applying α and β as shape and scale parameters, the gamma distribution can be expressed by probability density function (PDF) and the (CDF) as mentioned earlier (Equations (1) and (2)):

$$f(X) = \frac{1}{\beta^a \Gamma(a)} \chi^{a-1} exp\left(-\frac{\chi}{a}\right) \tag{1}$$

$$f(X) = \int_0^X f(t)dt \tag{2}$$

where χ represents the monthly average SM for the range $0 < \chi < \infty$ and $t$ illustrates a dummy variable. Later, the bias-corrected SMOS SM data was evaluated by the Nash–Sutcliffe efficiency (NSE) and root mean square ($R^2$). We used the Web-based-Hydrograph Analysis Tool (lit) for the current calculation (https://engineering.purdue.edu/mapserve/WHAT/cgi-bin/compute_r2_nash_sutcliffe.cgi). The tool is used in many cases of modeling to compute R² (Nash–Sutcliffe) coefficients to validate the model. This Web-based statistics module provides a tool for computation of these coefficients. The NSE value ranges from $-\infty$ to 1, and the $R^2$ value ranges from 0 to 1.

## 3. Results

### 3.1. Temporal SMOS Soil Moisture (SM) Analysis

We compared SMOS SM data to the in situ SM measurements, and examined their relation to climate (e.g., rainfall). Figure 4 shows the density scatter plots, which provide a quantitative comparison between the SMOS SM after bias correction and in situ SM measurements for the entire five year monthly SM mean value in Southwestern Mongolia (encompassing four different stations). The bias-corrected SM values are more consistent with in situ SM data than the original satellite SMOS SM products. The ratio method resulted in an $R^2 = 0.81$ in Bayankhongor, $R^2 = 0.77$ in Uvurkhangai, $R^2 = 0.74$ in Gobi-Altai, and $R^2 = 0.60$ in Arkhangai. The results suggest that SMOS SM data and precipitation time series show moderate compatibility. The monthly averaged SMOS SM data were strongly correlated with the average in situ SM measurements in the steppe and forest-steppe areas. These findings demonstrate that SM in these areas is relatively higher than SM in dry regions. In particular, 2012 was a relatively wet year in the provinces of Uvurkhangai and Arkhangai, when (on 19 July) SM reached a peak value of 16.8% and 15.9%, respectively. During the spring and summer, SMOS SM increased in May, reached a peak value in mid-July, and then began decreasing through the end of August. The consistency of this SM cycle between SMOS SM and in situ measured SM data implies that the absolute maximum reached at the end of July may be correlated with the vegetation development. However, the soil water that accumulated during the winter and spring precipitation events was sufficient for vegetation growth. Subsequently, the spatial distribution of SM depends on the soil parameters that were not distributed homogeneously in the study areas (soil texture, vegetation, and topography).

Figure 5 shows the density scatter plots that compare the SMOS SM data after the gamma distribution with in situ measurements. The gamma distribution algorithm successfully replicated in situ measurements. The bias-corrected SMOS SM data correlated well with the in situ SM data for all stations: Bayankhongor ($R^2 = 0.69$), Uvurkhangai ($R^2 = 0.83$), Gobi-Altai ($R^2 = 0.74$), and Arkhangai ($R^2 = 0.84$). Most of the in situ SM distribution was similar to the bias-corrected SMOS SM distribution, and thus met the primary conditions of the bias correction technique. According to Moriasi et al. (2007) the performance of the model is acceptable when the NSE and R² values are both greater than 0.5 [43]. Mostly, the greatest variations of SM were observed during October and April in both datasets. Assessment of gamma results indicate that, generally, strong correlations were noted for moist soils (Figure 5b,d), while lower correlations were noted for dry soils (Figure 5a,c). The results presented in Table 2 (as an example) show the summary of the gamma distribution algorithm applied in this study.

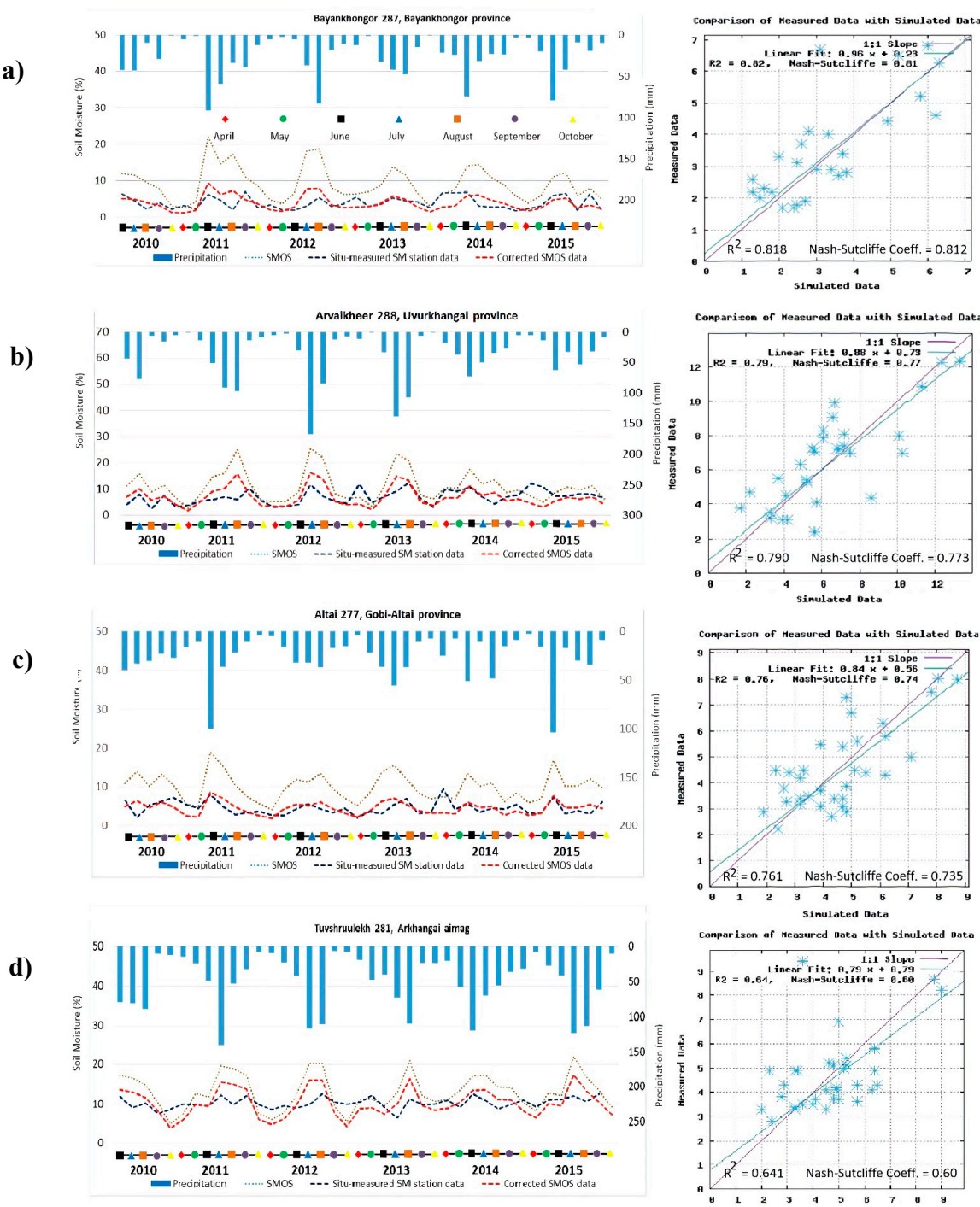

**Figure 4.** Result of ratio method and comparison of the in situ measurements SM data, and Soil Moisture and Ocean Salinity (SMOS) bias-corrected SM, and antecedent precipitation for (**a**) Bayankhongor, (**b**) Uvurkhangai, (**c**) Gobi-Altai, and (**d**) Arkhangai provinces.

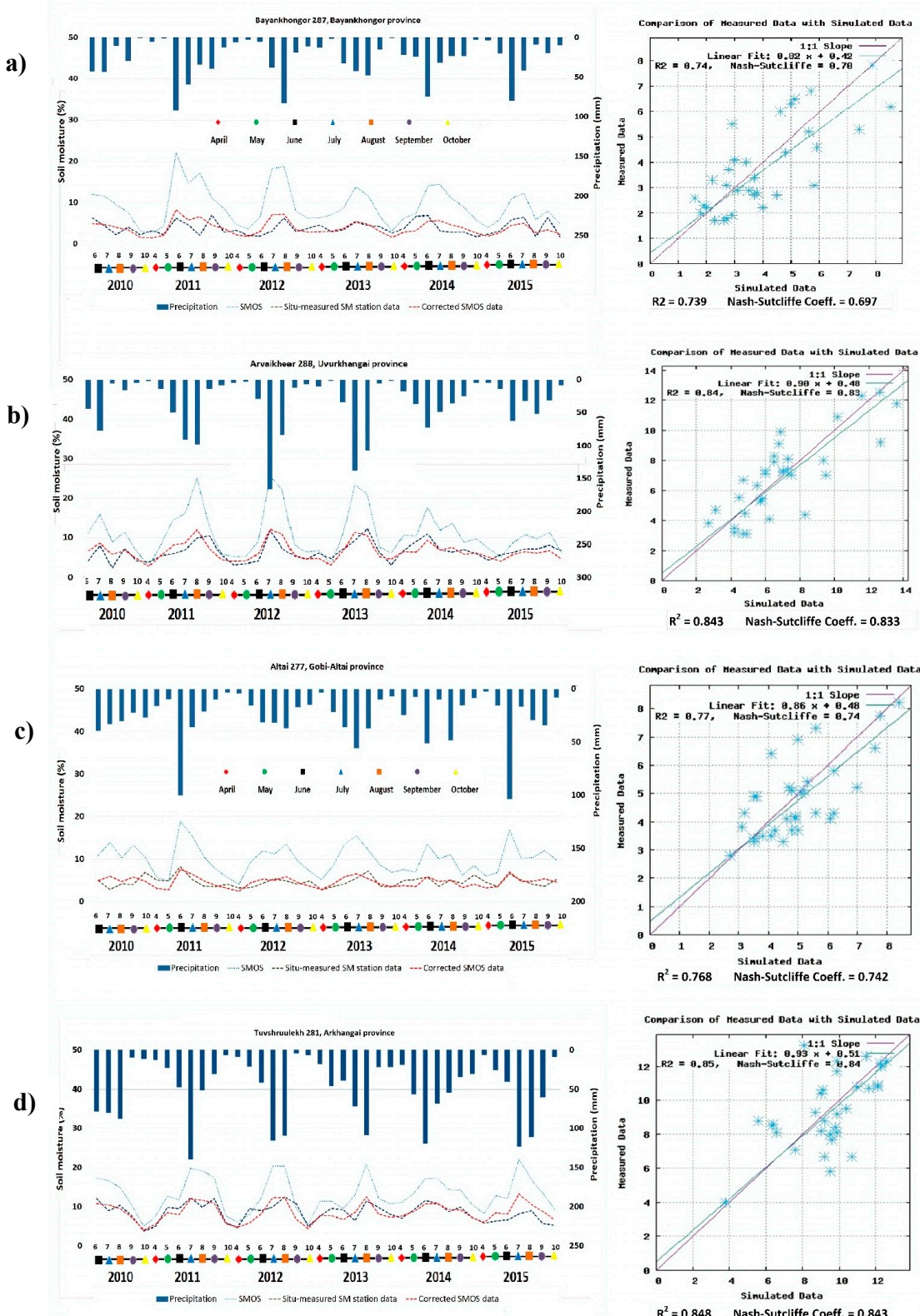

**Figure 5.** Result of gamma distribution method and comparison of the in situ measurements SM data, and SMOS bias-corrected SM and antecedent precipitation for (**a**) Bayankhongor, (**b**) Uvurkhangai, (**c**) Gobi-Atai, and (**d**) Arkhangai provinces.

**Table 2.** Summary of statistical parameters (gamma distribution algorithm) applied for bias correction of SMOS SM data in this study.

| Bias-Correction (Gamma Distribution Algorithm) Station 288/Arvaikheer | In Situ Measurements (Observation Data) | SMOS Data (Raw) | SMOS Data (Corrected) |
|---|---|---|---|
| Average moisture % | 6.68 | 11.05 | 6.69 |
| Standard Deviation | 2.49 | 5.85 | 2.46 |
| shape, alpha | 7.19 | 3.57 | 7.39 |
| scale, beta | 0.93 | 3.09 | 0.91 |

The bias-corrected SMOS SM data was applied to assess the grassland development status. Overall, the SMOS SM data show moderately good compatibility with expected relationships, in particular, increasing SM after rainfall. During the winter period, SMOS had no SM data signal because of snow, which produced outlier values at the beginning (April) and end of the periods (October).

### 3.2. Spatial Distribution of SMOS SM

We examined the spatial variability of SM data with bias-corrected SMOS SM data. Figure 6 shows the result of a corrected SMOS SM spatial distribution maps (absolute differences from 2010 to 2015). The maps indicate that the SM increased approximately 20% to 30% in 2011 in the northern part of Bayankhongor and northern central part of Uvurkhangai provinces, respectively. For 2012, we observed that SM increased by 2.5% to 10% for most of the investigated area. Hence, significant changes of SM can be seen in bias-corrected SMOS SM map Figure 6b,c. However, a subsequent SM distribution showed a significant SM decrease (30%) in the steppe and forest-steppe regions that contain Kastanozems haplic soils (Figure 6f). Further, bias-corrected SM distribution maps allowed delineation of wet areas in the northwestern and southeastern dry areas of Mongolia. Hence, the bias-corrected SMOS SM products imply that they could be effective for soil moisture monitoring.

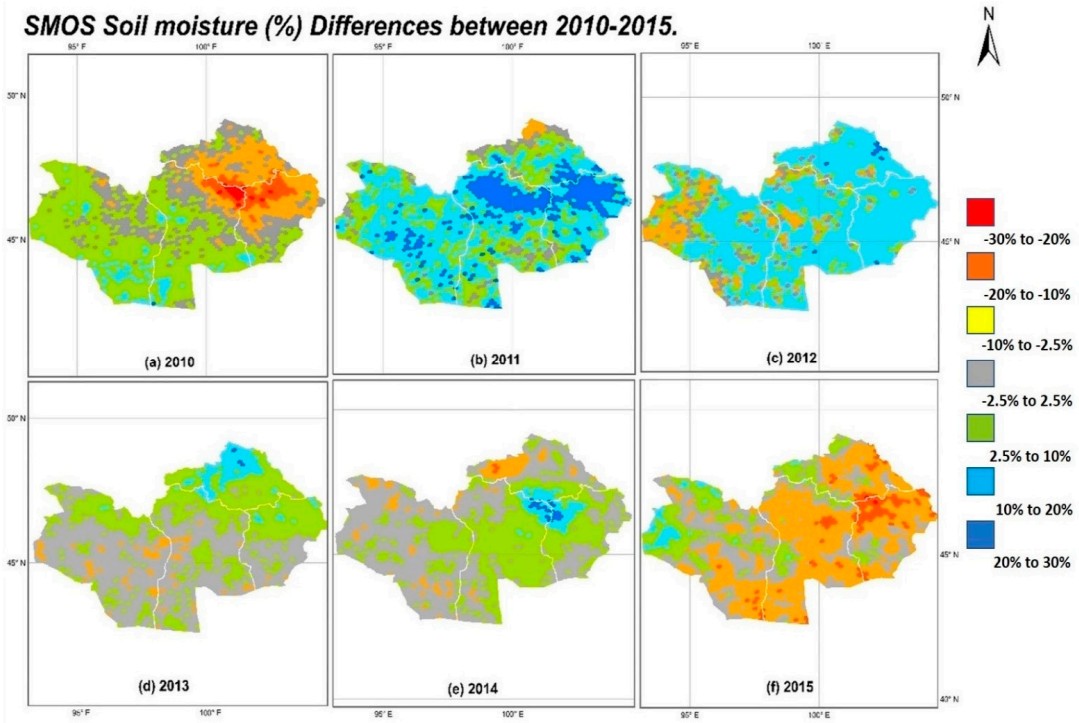

**Figure 6.** Spatial distribution of bias-corrected SMOS SM maps from 2010 to 2015 in Southwestern Mongolia.

### 3.3. The Relationship between SMOS SM and MODIS NDVI Vegetation

In order to check how NDVI represents the spatial differences in vegetation, we examined the relationship between SMOS SM and NDVI. Consistently, the NDVI is a proven indicator of vegetation, drought, and the thermal state of the land surface [44]. Figure 7 displays the relationship between SMOS SM (annual average) and MODIS NDVI (annual average) from 2010 to 2015. The results show that there is a positive tendency relating to the spatial patterns of NDVI. The bias-corrected SMOS SM favorably correlated with NDVI in the area around different soil types such as (Kastanozems haplic) with $R^2 = 0.94$ and $R^2 = 0.72$ (Figure 7b,d). Furthermore, lower correlations were found in semi-arid and desert regions (Chernozems calcic and Calcisols haplic) soils with $R^2 = 0.65$ and $R^2 = 0.62$, respectively (Figure 7a,c). The cause of these varying correlations is the different vegetation types. For instance, the southern region corresponds to open shrub-land, which is drier and has a smaller degree of vegetation content.

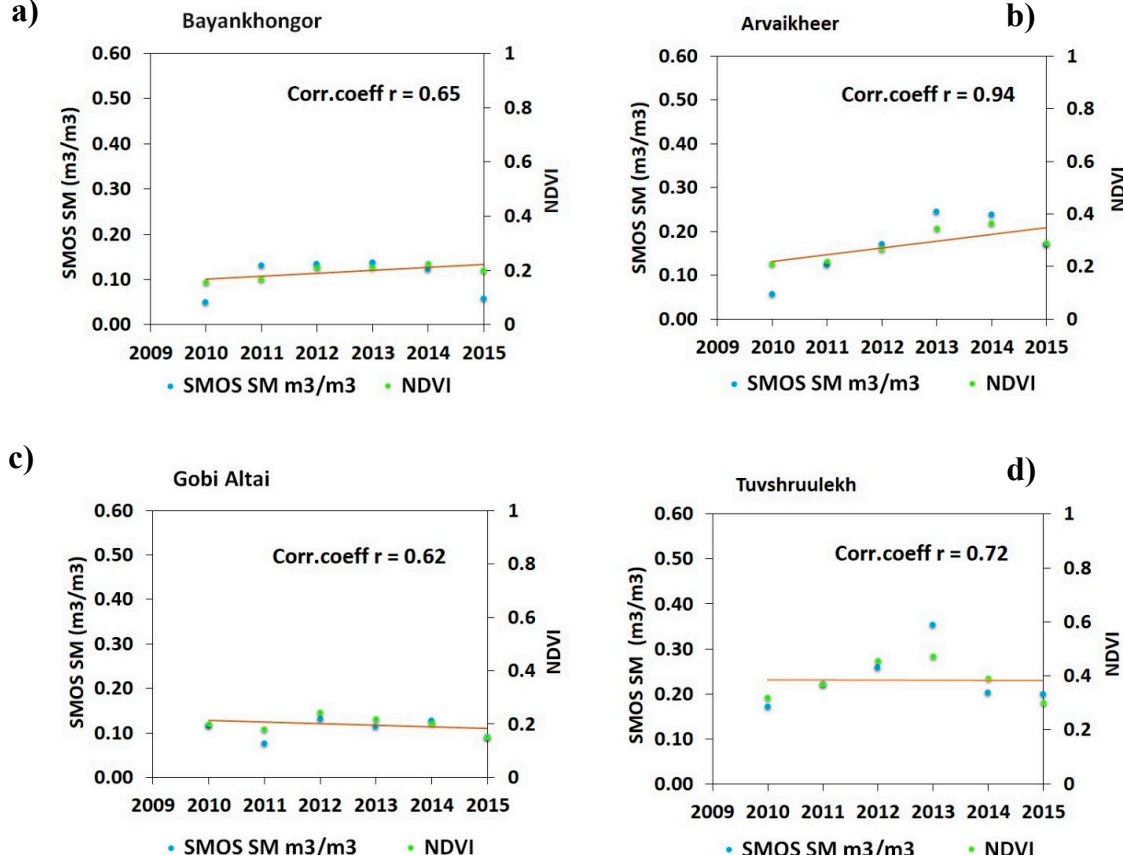

**Figure 7.** Scatter plot showing the annually-averaged correlation values obtained between SMOS SM and Moderate Resolution Imaging Spectroradiometer (MODIS) Normalized Difference Vegetation Index (NDVI) in the provinces: (**a**) Bayankhongor, (**b**) Arvaikheer, (**c**) Gobi-Altai, and (**d**) Tuvshruulekh.

Figure 8 presents the seasonal correlations between SMOS SM and NDVI. The relationship between SM and NDVI was examined using the seasonal SM values from April to July and July to October. From the regression plots, it can be seen that the seasonal values of NDVI and SMOS SM have the highest correlation during the growing season (Figure 8a,c,e,g). We detected a far weaker correlation for estimates during the non-growing season (Figure 8b,d,f,h). The seasonal NDVI values over the growing season correlate better in the humid and dense vegetation areas.

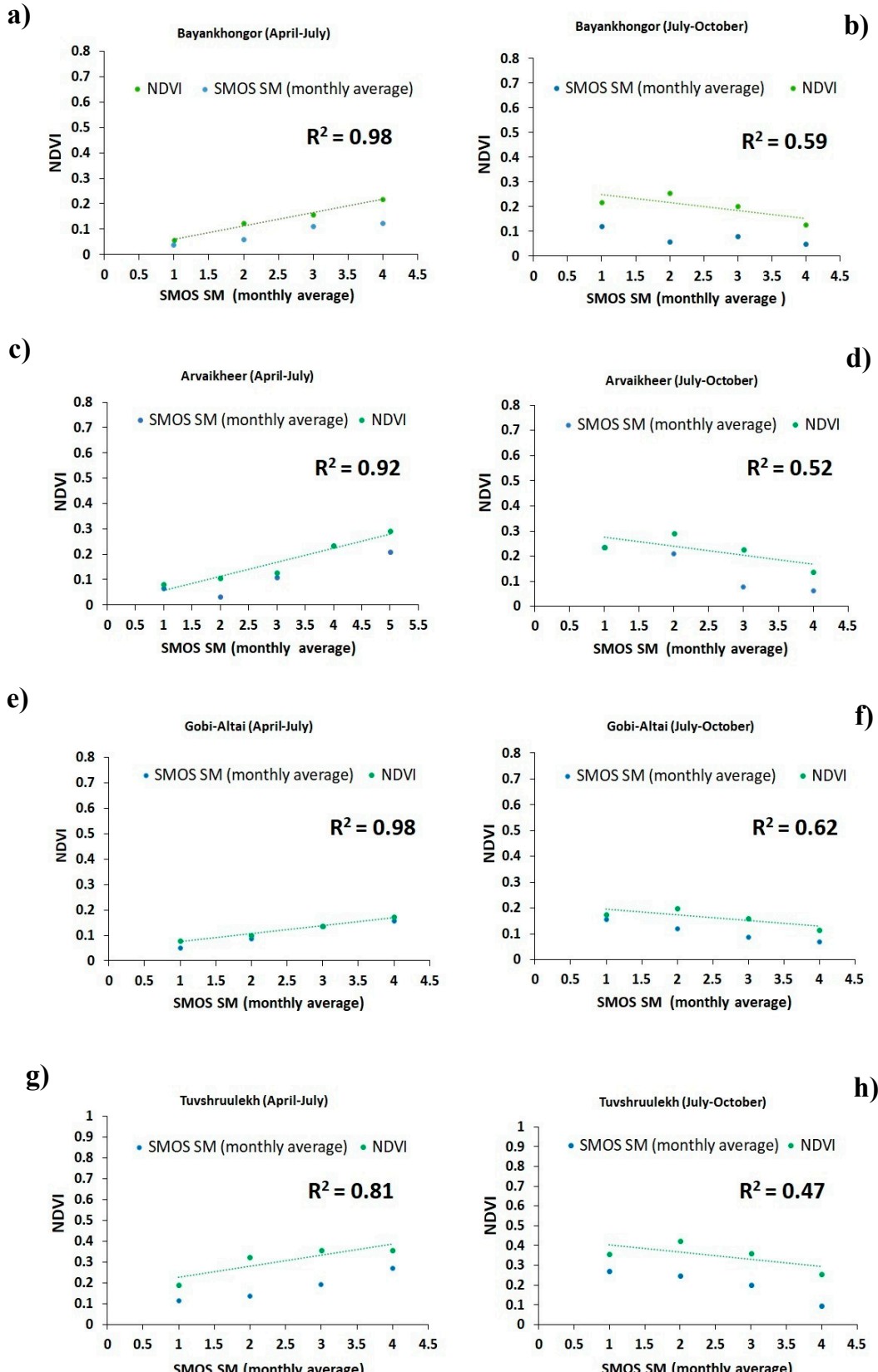

**Figure 8.** Scatter plot showing the monthly-averaged correlation values obtained from SMOS SM and MODIS NDVI during two seasons, April through June and July through October, in the provinces: (**a**,**b**) Bayankhongor, (**c**,**d**) Arvaikheer, (**e**,**f**) Gobi-Altai, and (**g**,**h**) Tuvshruulekh.

## 4. Discussion

### 4.1. Seasonal Precipitation and SM

The results reveal that the temporal variation and spatial distribution of the bias-corrected SMOS SM was generally related to precipitation, in agreement with other findings [33]. For the entire study area, peak SM was observed in late July and decreased in mid-August. The lowest SM was observed at the beginning of spring (i.e., April/May), which was due to the snow cover melting from the winter. Significant rainfall in July 2011 caused an increase in SM, which was recorded in both the bias-corrected SMOS SM data and the precipitation records at stations like Arkhangai (336 mm/year). In 2011 and 2012, the maximum SM (in July) depended strongly on the precipitation. In Gobi-Altai province (desert/high mountain region) the wettest year was 2011, however the highest SM value was detected in June, and actually decreased in July. The lower than average rainfall during this rainy season was the main cause of the SM reduction in late July. Subsequently, in dry years, we found a shift of the SM depression from August to mid-July. These findings could indicate that SM conditions during the early plant growth stage critically impact the vegetation condition, which is consistent with other studies [45]. When there is a lack of rainfall in drylands (Bayankhongor, Gobi-Altai), higher temperatures reduce SM, which is precipitation driven in these areas [46]. Moreover, low precipitation leads to SM deficits, increasing aridity and leading to drought. It is worth noting that many lakes in Mongolia have shrunk or dried up over the past decades [47]. In relatively dry areas covered by sparse vegetation, dry summer months also transport dust storm events causing additional damage related to, e.g., health, the environment, and the economy.

### 4.2. NDVI Vegetation Index and SM

Most of the bias-corrected SMOS SM distributions in this study were similar to NDVI and have similar dynamics. This is particularly the case in open shrub-lands, where NDVI values are low. Because of the widespread droughts, the dry regions are becoming drier and the growing season is getting shorter [48]. A previous study highlighted that data from satellite remote sensing data SMOS are strongly correlated with vegetation dynamics [49]. Bias-corrected SMOS SM and in situ SM were most correlated in humid areas (Figure 7). Considerably, the spatial distribution of SM depends on soil parameters that are not distributed homogenously in the area. This indicates, in general, that SM in dry steppe areas could change very rapidly in the top soil layer. The dynamics of NDVI show a good compatibility and strong correlation with the bias-corrected SMOS SM when measured in the seasonal cycles (Figure 8).

### 4.3. Relation of SMOS SM to Measured SM

The SMOS SM data strongly correlates with in situ SM measurements; the SMOS SM datasets successfully captured the spatiotemporal dynamics of the in situ SM measurements in the transition zones between dry and humid climates (as seen in the comparison analysis). Furthermore, for comparison analysis it is important to capture the correct temporal pattern of the in situ SM measurements. The SMOS data exhibited weaker correlations in some of the dry desert areas (e.g., Gobi desert), possibly caused by the relatively small range of SM values in these regions (which corresponds to remote sensing accuracy [19]). The desert and desert steppe regions are too dry to transmit an adequate signal to the sensor. With rising SM the quality of the signal may increase and allow a stronger statistical context in the northern regions. Some differences between SMOS SM and in situ SM measurements were observed, that may possibly be explained by the respective depths of the measurement; SMOS data is for depths from 0 cm to 5 cm, while the in situ measurements are between 10 cm and 15 cm [50]. Particularly, in the early spring and late autumn, the greatest variations of soil moisture were observed in both datasets. The SMOS data exhibited a certain underestimation in April and October compared to the ground observations, although the SMOS data reacted to rainfall events more quickly (Figures 4 and 5), which was also detected in previous validation experiments [51].

## 5. Conclusions

This study compared satellite SMOS SM with the in situ measured SM at depths between 10 cm and 15 cm from 2010 to 2015 in the southwest part of Mongolia. The spatial distribution of SMOS SM, MODIS NDVI and their relationship were used to assess the grassland condition. SMOS SM was also compared with in situ SM measurements to examine the reliability of SMOS SM. Two techniques (i.e., ratio and gamma) were applied to correct the bias between the in situ SM and SMOS SM data. The in situ measured SM distribution was close to the bias-corrected SMOS SM distribution, and thus met the primary condition of the bias-correction technique. The two algorithms utilized for comparison analysis successfully recreated the in situ measurements. For all investigated stations, the coefficient of determination ($R^2$) ranged from 0.6 to 0.8 for the validation results. The distribution patterns of bias-corrected SMOS SM correctly reproduced the precipitation season from June to July as well as the drying out period starting in October. In both datasets, the highest variability of SM occurred in the northern part of the study area in 2011, with an increase in SM of up to 30% above 2010 levels. The lowest SM values were observed in 2015, with significant decrease (30% compared to 2010) in the steppe and forest-steppe regions with Kastanozems haplic soils. Overall, the small seasonal changes in the bias-corrected SMOS SM and in situ measurements were generally similar throughout the study area during the three phases of observed vegetation growth (i.e., warm spring, summer recharging, and autumn drying season). The study site lies in a zone that transitions between steppe, forest-steppe, and desert. The lowest correction between SMOS SM and in situ SM was observed in the dry, lowland regions. While the spatial resolution of SMOS is coarse, the high temporal resolution of the SMOS SM data will be useful to determine large scale temporal SM changes.

This study confirms a simple bias-correction technique (ratio and gamma distribution) is a valid method to compare in situ SM measurements and remotely sensed SMOS SM data in the southwestern part of Mongolia. The use of SMOS SM data allows easier monitoring of spatial and temporal changes. The geostatistical results and spatial distribution SM maps will be useful in grassland development studies for the purpose of addressing drought in Mongolia. In the future, the bias-corrected SMOS SM products can be used as an extra tool for monitoring the grasslands of Mongolia.

**Author Contributions:** O.V. designed the study, interpreted the results, provided resources for the literature review and wrote the manuscript. Authors M.K. and A.R.E. edited and provided conceptual contributions and reviewed the manuscript drafts. All authors read and approved the final manuscript.

**Funding:** This research was funded by the Areas Plus project of European Commission, grant number EU-1606.

**Acknowledgments:** The study was supported by the Areas Plus project of European Commission, (Grant No: EU-1606) at the Department of Cartography, GIS and Remote Sensing of Göttingen University. We are grateful to the National Agency for Meteorology, Hydrology and Environment Monitoring of Mongolia (NAMHEM) for providing data. The authors would like to thank the European Space Agency (ESA; SMOS Expert Support Laboratory) for providing SMOS products respectively.

**Conflicts of Interest:** The authors declare that they have no competing interests.

## Abbreviations

| | |
|---|---|
| NDVI | Normalized Difference Vegetation Index |
| EVI | Enhanced Vegetation Index |
| SAVI | Soil-Adjusted Vegetation Index |
| SMOS | Soil Moisture and Ocean Salinity |
| SSMR | Scanning Multi-channel Microwave Radiometer |
| SSM/I/DMSP | Special Sensor Microwave Imager/Defense Meteorological Satellite Program |
| TMI/TRMM | Microwave Imager/Tropical Rainfall Measuring Mission |
| AMSR-E/EOS PM-1 | Advanced Microwave Scanning Radiometer – EOS |
| MIRAS/SMOS | Microwave Imaging Radiometer using Aperture Synthesis/Soil Moisture Ocean Salinity |
| Aquarius/SAC-D | Aquarius/Satélite de Aplicaciones Científicas-D |
| SMAP | Soil Moisture Active Passive |

| MARCC | Mongolia Assessment Report on Climate Change |
|---|---|
| ECV | Essential Climatic Variable |
| NAMHE | National Agency of Meteorology, Hydrology and Environment Monitoring of Mongolia |
| LP DAAC | U.S Land Processes Distributed Active Archive Center |
| CDF | Cumulative Distribution Function |
| NS | Nash–Sutcliffe |
| PDF | Probability Density Function |

## Appendix A

**Table A1.** Summary and general characteristics of SM sensors in the last and current decade.

| Sensor/Space Mission | Operation Period | Frequency (GHz) | Resolution (km) | Incidence Angle (°) |
|---|---|---|---|---|
| **SSMR/Nimbus-7** | 1978–1988 | 6.6, 10.7, 18, 21, 37 | 150 (at 6.6 GHz) | 50 |
| **SSM/I/DMSP** | 1987- | 19.3, 22.3, 37, 85.5 | 25 (at 19.3 GHz) | 53 |
| **TMI/TRMM** | 1997–2015 | 10.65, 19.35, 21.3, 37, 85.5 | 50 (at 10.65 GHz) | 53 |
| **AMSR-E/EOS PM-1** | 2002–2011 | 6.9, 10.7, 18.7, 23.8, 36.5, 89 | 56 (at 6.9 GHz) | 55 |
| **AMSR2/GCOM-W1** | 2012 | 6.92, 7.3, 10.65, 18.7, 23.8, 36.5, 89 | 10 (at 6.92 GHz) | 55 |
| **MIRAS/SMOS** | 2009 | 1.42 | 25 to 40 | 0 to 55 |
| **Aquarius/SAC-D** | 2011–2015 | 1.41 | 76×94, 84×120, 96×156 | 29, 38, 46 |
| **SMAP/SMAP** | 2015- | 1.43 | 40 | 40 |

Source: [52], SSMR = Scanning Multi-channel Microwave Radiometer; SSM/I/DMSP = Special Sensor Microwave Imager/Defense Meteorological Satellite Program; TMI/TRMM = Microwave Imager/Tropical Rainfall Measuring Mission; AMSR-E/EOS PM-1 = Advanced Microwave Scanning Radiometer – EOS; MIRAS/SMOS = Microwave Imaging Radiometer using Aperture Synthesis/Soil Moisture Ocean Salinity; Aquarius/SAC-D = Aquarius/Satélite de Aplicaciones Científicas-D; SMAP = Soil Moisture Active Passive.

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
