# Peer review of "Comparison of Satellite Soil Moisture Products in Mongolia and Their Relation to Grassland Condition"

_land, doi:10.3390/land8090142_

Round 1

Reviewer 1 Report

This manuscript has improved considerably since its initial version. I thank the authors for having accounted for all the reviewers' comments, including mine. The study interest is better conveyed now.

I had the impression that the authors had worked hard to improve the manuscript, and in the process, several small aspects escaped their attention. This justifies my recommendation of returning the manuscript to them for a minor revision. However, please take the notes below as suggestions the authors might like to revise, rather as sharp conditions to be met before publication. Most of the notes below concern formal points rather than scientific contents.

Title: Specifying which products might increase the manuscript reach. 'for' or 'in' Mongolia could be better than 'of', but this should be assessed by a native English speaker. 'Condition' could be more specific than 'development'.

l 3: As soil moisture may also have a positive effect on agriculture and grasslands, a more precise statement would substitute 'degradation' by 'condition'.

l 7: Remove initial period.

l 19: Please rephrase or include reference.

l 37: Please start with capitals and rephrase. It looks like a loose line coming from elsewhere.

l 52: This range of temperatures seems to be wrong (I know it is not), or a variation of 1 ºC does not justify the term 'range'.

l 55: Perhaps 'm.a.s.l.' would avoid a repetition.

Figure 1. The upgraded Figure is very good. Please clarify the origin of the (eco)regions.

Figure 2: Again, please say the origin of this map.

Figure 3: A flowchart should have unambiguous and unidirectional connectivity between boxes. This is not the case of 'Bias correction'. I personally found the earlier version slightly easier to follow.

l 195: This way to introduce references, where their number (here, '[39]') is used as a name, appears repeatedly in the text and is somewhat weird. I would suggest making the statement and put the reference number, or explicitly mentioning the authors and then using the number instead the year.

Appendix 1: the column 'Resolution (km)' offers inconsistent entries with different formats: '150 at 6.6', '25-40', '76*94', '40'. Please unify and explain clearly the interpretation. I suggest to use 'by' or 'x' instead '*'.

Author Response

Dear Reviewer,

We would like to thank you for the opportunity to revise our manuscript, “Comparison of satellite soil moisture products in Mongolia and their relation to grassland condition” 592343. We would also like to take this opportunity to express our thanks for the positive feedback and helpful comments for correction and modification. It is our belief that the manuscript is substantially improved after making the suggested edits.

Thank you very much for your kind consideration.

Sincerely yours,

Oyudari Vova

Reviewer 2 Report

The study is very important in monitoring soil water resources in the context of climate change and the impact on vegetation.
The paper was radically improved by the recommendations of the reviewers.
In this situation I consider that the paper can be accepted for publication in this form.

Author Response

Dear Reviewers,

Thank you very much for reviewing our manuscript. We also greatly appreciate the reviewers for their complimentary comments and suggestions. We have carried out the manuscript that the reviewers suggested and revised the manuscript accordingly.

Thank you very much for your kind consideration.

Sincerely yours,

Oyudari Vova

This manuscript is a resubmission of an earlier submission. The following is a list of the peer review reports and author responses from that submission.

Round 1

Reviewer 1 Report

The topic of this study is vey good. Land surface soil moisture condition is an important factor influencing vegetation growth in the arid or semi-arid regions. However, through this study, the analysis is very basic and cannot provide the right answers to the topic. Because the analysis is simply based on four stations with few analysis at the regional scale. Both smos data and modis data could provide regional information of the study area. The authors didn’t use them well. In addition, the big spatial difference between the station measurements and the SMOS observation is also an important issue should be discussed in the manuscript. However to handle this problem should be mentioned in the study. Moreover, the organization and the expression of the manuscript is a little confused that the readers are hard to get the full purpose. Therefore, I think is should be rejected in current version and it will be encouraged to be resubmitted after major revision.

Comment:

1.       Line 11-12: identify vegetation?

2.       Line 15-20: I do not know why the authors suddently mention SMOS soil moisture product?

3.       Line25-27: why introduce all the soil moisture satellite sensors here?

4.       Line 35, please provide the full name of SM when first use it. Check the similar problem throughout the manuscript.

5.       Line 37-29, I think this sentence has not connection with the above sentence.

6.       Line 48-50, there is no relationthip for this sentence with the above ones.

7.       In general, I think the introduction section is totally confused without logical relationship. The authors should make a big revision to this part to present a clearer introduction to the status and the purpose of this study. What is the major problem you want to solve or discuss during the analysis.

8.       Line 64-65, which province? There is no such information in the above sentence.

9.       For the introduction of the study area, I think it is better for the authors to introduce the information of the study area as a entirety but not separated into four province. In addition, the mountains and land cover pattern of the study area should be depicted in Figure 1.

10.   Figure 1: it is hard to figure out the name of the province.

11.   Line 122-124: Why only three times every month? I remember that the SMOS SM could provide daily SM observation.

12.    Line 136-138: Why use MOD11A1 and MYD11A1 land surface temperature product? I remember that there is no work related to this parameter.

13.   Line 158-159: what is the unit difference? Which unit the SMOS data converted to?

14.   The subfigures in Figure 5 should be numbered with a new series not from previous figure 4.

15.   Line 232-234: How to get this conclusion. Please use some metrics to specify the result.

16.   Figure 6: Because the figures are the difference maps. Therefore, how to get the results of the first year?

17.   For figure 7 and 8, I do not know how to get the fitted line?

18.    

Reviewer 2 Report

Review for the paper entitled
“Monitoring of soil moisture in Mongolia and its relation to grassland degradation”
By Oyudari Vova, Martin Kappas and Ammar Rafiei Emmam
Manuscript ID: land-498827
Recommendation: Major revisions.
This study analyses the soil moisture from the SMOS satellite in the southeastern part of Mongolia (divided in 4 provinces). The first part aims at correcting the bias of the satellite measurements through two simple methods and comparing with in situ measurements from 4 different sites. The second part analyses the evolution of this soil moisture measurements (SMOS and in situ) from 2010 to 2015 into the 4 provinces, linking it with the rainfall. This part also evaluates the two simple methods to remove the SMOS soil moisture bias, in order to be able to compare the satellite data with the observations. The third part analyses the spatial distribution of the soil moisture in the region along these years and, finally, this is linked with the NDVI to try to relate the evolution of soil moisture with the grassland degradation. The motivation of the article is good, the strategy is correct and the results are interesting (despite some of them were expected). However, the paper lacks in clarity along the text, consecutive sentences are not correct and the organization of the article should be improved significantly. There are also some details of the analysis that should be reconsidered and some conclusions need to be better demonstrated. Besides, in some cases, the figures and the text are hard to follow. I propose below some major changes and other minor comments that could be applied before the manuscript is accepted in the journal. I also recommend revising exhaustively the English in the whole manuscript, since several sentences should be reworded in an appropriate way. I hope my comments can help to improve this manuscript and to allow its final publication in this journal.
Major comments
1. The structure of the paper could be improved. The subsections are not completely clear and in many cases is a little bit hard to follow. The paper has three main parts: 1. Bias correction of SMOS data. 2. Rainfall and soil moisture analysis. 3. Spatial analysis and relation with NDVI.
2. Section 2 is also a little bit confusing with too many subsections
3. The introduction Section could be improved to better demonstrate the motivation of this work. I also think that some parts of the Introduction should go to Section 2 (especially the discussion about SMOS SM L2).
4. Why do you use SMOS SM L2 and not L3, such as the product provided by CATDS?
5. The figures can be improved in quality and the caption of the figures should express more clearly what it is shown in them.
6. The discussion Section is quite difficult to follow. I think this section is needed in this paper, but the discussion should be re-written in a clearer way, especially trying to better connect consecutive sentences, which in many cases seem disconnected.
7. Even when I am not an English reviewer, I think that the English language should be revised by a native. There are some sentences that are difficult to understand. I think an effort should be made to improve the language. I provide only a few English comments before, but there are more.
Minor comments
- I think Table 1 is not needed in the paper.
- Lines 35-37 and 37-41 are not connected between them
- Please, provide a better motivation for this work at the end of the Introduction. The motivation is given before, but it would be good to have it at the end of the Introduction.
- L. 52 – “to investigate” instead of “to access”.
- L. 60 – Maybe it is better to change the colorbar of the figure to these values of height.
- L. 64 – In line 60 you said 2414 m maximum.
- Please, in section 2.1, provide a table with the main information of each province, maybe adding: Province-Temp-Precip-Soil-Land cover. Try to organize better the text to be clear of which province you are talking about.
- Figure 1 – Please, improve the figure caption. The names of the provinces are hard to read. Please, add the dates of the soil moisture map. Do you think that the soil moisture data in the mountainous region are OK? Add the name of the in situ stations in the map, as well as different sites that you mention in the text, including provinces names. Add panels’ numeration (a, b, c). The same can be applied to all the figures.
- L. 93 “In situ”
- L 101 – “at two vertical depths”
- L 101 – Do you mean 0-15 cm or 15 cm?
- Figure 2 – Maybe it would be nice to add a map of land cover, or both, land cover and soil type. Where does this map come from?
- L 105 – “The ground-based soil-moisture measurements”
- L. 110 – “usefulness” by “reliability”
- Figure 3. This conceptual figure is too long, please, try to do it more concise.
- L. 118 – I do not think that “data assimilation” is the most appropriate word for your work. I think you should change this “assimilation” by “pre-processing” or “processing” in the whole text.
- L. 120-121 – Please reword this sentence.
- L. 123 – Why do you only have sm measurements three times per month?
- L 128 – “The measured station SM data” sounds very strange.
- L. 136 – “The NDVI data from MODIS used…”
- L. 138 – “…respectively). These products…”
- L 147 – Please, add a space before all units in the whole text (1000 mm).
- L.158-159 – I think this sentence is out.
- L. 161 – “The accurate data assimilation systems…” I do not know the meaning of this sentence.
- Section 2.2.2.1 – This is a part of your work, could you maybe try to explain better each of the two methods that you use?
- L. 177-178 – Please, improve the quality of the formulas.
- L. 181 – Could you better explain what is Nash-Sutcliffe efficiency (NSE)?
- L. 190 – Why the correlation is lower in Arkhangai province? Could
- L. 192 - You say 2011 and 2012, but the analysis is done for other years, could you analyse a little in the text the other years?
- L. 193 – “stronger”
- L. 193-194 – Why does this mean higher soil moisture?
- L. 200-201 – Or maybe the maximum in vegetation development is related to the soil moisture?
- Figure 4 –
o Where do the rainfall data come from?
o The sm values are not higher than 0.1 in many cases, maybe you could change the y axis of soil moisture.
o Why the dates are only given from April to October?
o It would be nice to have a table with annual rainfall at the four sites, or maybe cumulative rainfall figures. This can help the reader to have a better picture of the wetness of each year.
o Please, in the scatter plots, change “simulated” by “corrected”
o Maybe you could maintain the y-axis range for the rainfall in the four figures, it would be better to compare the four sites, although the table or the cumulative rainfall figure can be better to this aim.
o Looking at the figure 4d, I do not see a lower correlation between in situ and satellite data than for the other provinces. Is this correlation well calculated? I see some values above 10 % in the time series but not in the scatter plot.
- Figure 5
o Why only two figures are shown and not 4 like in Figure 4.
o Please, change “e” and “f” in the figure.
- L. 220-222 – Sentence not clear.
- Table 3 – Please, explain before the coefficients shape, alfa and scale, beta.
- L. 231 – “Subsequently, the bias…”
- L. 235 – “The soil moisture captured the peak rainfall” – Remove this sentence.
- Figure 6
o Please, add detailed info about the figure.
o The figure shows absolute differences between what? It is not clear, please clarify.
- L. 258 – It would be better to analyse this relationship using monthly data.
- Figure 7
o Please change y-axis to more appropriate values (0-0.2 for example).
o What are the colors in the legend?
- Figure 8.
o Please, improve the quality of the figures
o Are these plots done with data from all years? I only see four points in each figure. Please, clarify.
o Please, improve the figure caption, since it is not clear like it is now.
o Figure 8h  I do see a quite clear correlation between the 4 points, I do not understand why it is only 0.20.
o Maybe July-August could be included with April-July, since there is still a growth of the plant.
- L. 284-285 – “long-term spatial variation” I do not see this, do you use all years?
- L. 289 – I do see a good correlation, please check.
- L. 290-291 – How the averaging would improve the relationship between variables? You do not find the same relationship without averaging?
- L. 294-300 – I think this should be in the introduction section.
- L. 300-302 – I think the decrease in sm (and rainfall) was only observed in 2015, so, you cannot attribute this to a trend, I think it was a dry punctual year.
- L. 306-315 – These sentences are not linked between them, they are not connected and the discussion becomes unclear. Please, reword trying to keep a continuous discussion.
- L. 318-320 – The layer of the soil is always shallow, not only when it is wet, isn’t? Please, clarify.
- L.322-324 – How do you extract this conclusion about the land cover if you have not analysed it here? Please, clarify.
- L. 335 – Why Arkhangai province has the wettest year in 2012 and not the others?
- L. 339 “could explain”
- L.339-340 – Talk before a little bit about the NDVI analysis.
- L. 341-343 – This last sentence is not clear.
- L. 345 – Remove “vegetation”
- L. 352-356 – These sentences are repetitive.
- L. 356-358 – This sentence is unclear.
- L. 365 – Why does this relationship depend on topographic relief?
- L. 366-367 – Please, reword.
- L. 368 – “The more significant…”
- L. 369 – “The less significant…”
- L. 369 – Why the more insignificant relationship was found in the lowland driest regions?
- L. 372-372 – How will this study contribute to enhance the spatial resolution of SMOS? I think you want to say another thing.
- L. 373-376 – “This study shows how the simple bias correction method is suitable…” “It also analyses the soil moisture spatial distribution…” Please reword these sentences.
- Could you maybe connect the final conclusions with some of the previous findings in the region?

Reviewer 3 Report

This is a valuable study on analyzing correspondence between EO-detected and ground-measured soil moisture. It is applied on a large study area of a high scientific and natural interest. In general terms, the deploy of analysis methods is appropriate and carefully developed. However, in my opinion it fails on some of the underlying assumptions and therefore on the yielded conclusions. In addition to that, the presentation of results is unnecessary long and delves into many regional details that do not affect the main conclusions and makes it difficult to follow the study structure. I would personally appreciate that such descriptions were moved to an appendix, or their inclusion in the main text were better justified.

Such problems should be reviewed by the authors, possibly requiring important rewriting of the manuscript. This justifies my recommendation to reconsider after a major revision.

Please see below some concrete observations justifying the said aspects:

Title: Whilst the overall quality of the English is satisfactory (but still requiring wordproofing), several spell or grammar errors have occurred along the manuscript. The title is a good example.

Table 1: The much-needed information on spatial resolution seems wrongly conveyed in the corresponding column (which uses GHz as units).

l. 38: It is not clear why transitions in soil composition favour higher evapotranspiration.

l. 40: Degradation is normally defined as resulting from human activities, not from abiotic drivers. While the former is normally beyond ecosystem resilience, the stress associated with the latter trend to favour adaptation. The fact that vegetation becomes greener in wet conditions, and less covering in dry conditions, suggests adaptation and resilience, rather than degradation. This conceptual mistake underlies later interpretations of results in the discussion.

l. 88: Further, the authors are aware of the point above. The statement here is contrary to the study basis that a low soil moisture causes vegetation 'degradation'. Thus I suggest a whole assessment of the term 'degradation' throughout the manuscript.

l. 112: This should be further justified. It is evident that soil moisture and vegetation cover will be positively correlated. But, 1) lags or delays between SM and NDVI may convert the latter in an unreliable estimator of the former for validation purposes: and 2) the fact that NDVI is low when SM is low does not mean that the corresponding ecosystem is degraded, but that it is resilient.

Section 2.2.1.1: Please clarify the spatial resolution of SMOS L2, and how any difference with the resolution of MODIS NDVI at 1 km was accounted for.

Section 2.2.1.2: Please specify the sampling design of measured SM, its spatial pattern and its relationship with the SMOS L2 resolution. This is very important, as normally ground-measured data show a higher variance than equivalent EO data.

Section 2.2.2.1: A short explanation of how bias work in SMOS data would be necessary.

l. 311: Just a question: could negative correlation between SMOS and NDVI occur in heat-limited rather that water-limited environments (e.g. mountains)?